# Lignin Extraction from Waste Pine Sawdust Using a Biomass Derived Binary Solvent System

**DOI:** 10.3390/polym13071090

**Published:** 2021-03-30

**Authors:** Solange Magalhães, Alexandra Filipe, Elodie Melro, Catarina Fernandes, Carla Vitorino, Luís Alves, Anabela Romano, Maria G. Rasteiro, Bruno Medronho

**Affiliations:** 1Department of Chemical Engineering, CIEPQPF, University of Coimbra, Pólo II–R. Silvio Lima, 3030-790 Coimbra, Portugal; solangemagalhaes@eq.uc.pt (S.M.); alexandraf@eq.uc.pt (A.F.); mgr@eq.uc.pt (M.G.R.); 2Department of Chemistry, CQC, University of Coimbra, Rua Larga, 3004-535 Coimbra, Portugal; elodie.melro@uc.pt (E.M.); csfernandes@student.uc.pt (C.F.); csvitorino@ff.uc.pt (C.V.); 3Faculty of Pharmacy, University of Coimbra, 3000-548 Coimbra, Portugal; 4Centre for Neurosciences and Cell Biology (CNC), Faculty of Medicine, University of Coimbra, 3004-504 Coimbra, Portugal; 5MED—Mediterranean Institute for Agriculture, Environment and Development, Faculdade de Ciências e Tecnologia, Universidade do Algarve, Campus de Gambelas Ed. 8, 8005-139 Faro, Portugal; aromano@ualg.pt (A.R.); bfmedronho@ualg.pt (B.M.); 6Surface and Colloid Engineering, FSCN, Mid Sweden University, SE-851 70 Sundsvall, Sweden

**Keywords:** biomass fractionation, formic acid, levulinic acid, lignin, maritime pine

## Abstract

Lignocellulosic biomass fractionation is typically performed using methods that are somehow harsh to the environment, such as in the case of kraft pulping. In recent years, the development of new sustainable and environmentally friendly alternatives has grown significantly. Among the developed systems, bio-based solvents emerge as promising alternatives for biomass processing. Therefore, in the present work, the bio-based and renewable chemicals, levulinic acid (LA) and formic acid (FA), were combined to fractionate lignocellulosic waste (i.e., maritime pine sawdust) and isolate lignin. Different parameters, such as LA:FA ratio, temperature, and extraction time, were optimized to boost the yield and purity of extracted lignin. The LA:FA ratio was found to be crucial regarding the superior lignin extraction from the waste biomass. Moreover, the increase in temperature and extraction time enhances the amount of extracted residue but compromises the lignin purity and reduces its molecular weight. The electron microscopy images revealed that biomass samples suffer significant structural and morphological changes, which further suggests the suitability of the newly developed bio-fractionation process. The same was concluded by the FTIR analysis, in which no remaining lignin was detected in the cellulose-rich fraction. Overall, the novel combination of bio-sourced FA and LA has shown to be a very promising system for lignin extraction with high purity from biomass waste, thus contributing to extend the opportunities of lignin manipulation and valorization into novel added-value biomaterials.

## 1. Introduction

Lignocellulosic biomass valorization for potential use in the production of biochemicals, biofuels, biomaterials, and other added-value products, represents an important opportunity to reduce and valorize agroforest residues [1,2]. In this respect, an important contribution can come from the pulping industry where considerable amounts of lignin-rich fractions are still poorly explored and valorized, despite their potential as a natural source of polyphenols [3]. Typically, such pulping approaches target cellulose isolation via somewhat environmentally harmful processes, such as kraft. Despite the major scientific and technical advancements and investments to reduce the environmental impact, the development of novel “green” biomass fractionation methodologies has been gaining growing attention from industries and governments. This shift in behavior has been triggered particularly by the sustainability goals established by the United Nations agenda [4]. Due to the high variety and heterogeneity of biomass raw materials and the complex hierarchical arrangement and interactions among the different biomacromolecules (such as cellulose, lignin, and hemicellulose [1,5]), fractionation and isolation are still a great challenge. Of particular interest is lignin, the second most abundant biopolymer in the world, constituting ca. 25% of the terrestrial plant biomass [6]. It is an inexpensive and renewable resource that possesses numerous attractive properties, such as high stability, biodegradability, and antioxidant activity [7]. All these attractive features, make lignin a very interesting raw-material to be applied in different areas, such as biomedical and pharmaceutical applications, resins, biofoams, or food packaging [8]. This biopolymer is water insoluble and stable in nature, and acts as the “glue” that connects cellulose and hemicellulose in the fibers of the plant cell walls [1,9,10,11]. Furthermore, since lignin is a highly branched amorphous polymer composed of phenol derivatives, it is regarded as an important bio-renewable source of aromatic compounds [10]. Despite its huge potential, most of it is still discarded or burned as fuel for energy production, and only ca. 1–2% of lignin has been, so far, utilized as a high value product [10,12]. 

Numerous physical, thermal and chemical pre-treatments and their combinations have been developed to boost biomass fractionation and isolation [13,14]. These pre-treatments of the lignocellulosic biomass are crucial for its efficient valorization. Nevertheless, such approaches are typically cost-intensive, accounting for up to 40% of the total processing expenses [15]. From the available pre-treatments for lignocellulose fractionation and lignin isolation, ethanol organosolv has been shown to be the most effective method for extracting lignin from pine [16]. Ethanol organosolv is referred to as a selective pre-treatment able to hydrolyze the internal bonds in lignins as well as lignin–hemicellulose bonds, contrary to the acidic hydrolysis conditions, which easily hydrolyses α-ether linkages (lignin), but it is likely that β-aryl ether bonds (cellulose) are also broken under the conditions used in many organosolv-based processes [17]. Another interesting system used in pulping is formic acid (FA), which is a by-product from lignocellulosic biomass processing [18].

As an environmentally benign and green solvent, water is most always preferred and has been extensively utilized for biomass fractionation and chemical production [19]. However, it has been demonstrated that water-based systems (i.e., subcritical water) can cause excessive and non-selective degradation if too high temperature and long residence times are employed [20]. 

Recent studies demonstrate the suitability of eutectic solvents and ionic liquids for biomass fractionation [21,22,23,24,25]. These systems are reasonably easy to prepare in a pure state, do not require the presence of any solvent, and produce no waste. In this respect, mixtures of bio-sourced cations and anions look very attractive.

Despite the recent efforts to improve lignin isolation and purification, it is clear that the vast majority of lignin is still obtained as a low-quality byproduct from pulping industries. To continue boosting the lignin-based materials and chemicals expansion towards novel advanced and demanding applications, superior fractionation strategies are required to improve lignin quality and its selective extraction. Therefore, in this work, using a closed reactor, capable of handling high pressures, a novel binary mixture composed of LA and FA is suggested as an efficient medium for lignocellulosic biomass fractionation. 

LA has been identified as a sustainable platform and building block for many valuable chemicals [26]. This acid can be obtained from fructose or cellulose, and is catalyzed by homogeneous or heterogeneous catalysts, such as phosphotungstic acid [27]. Also, FA pulping is one of the most promising organosolv-based fractionation chemicals, allowing an efficient fractionation of the raw biomass into a cellulose-rich fraction, a water-soluble fraction rich in sugars, and a lignin fraction [25]. FA can be easily recovered by distillation [28]. 

The fractionation suitability of this novel binary system was studied, and the extraction conditions, such as the LA:FA ratio, temperature, and extraction times, were optimized to maximize the lignin yield and purity. The extracted lignin was further characterized by Fourier-transform infrared spectroscopy (FTIR) and scanning electron microscopy (SEM).

## 2. Materials and Methods

### 2.1. Materials

Maritime pine (*Pinus pinaster* Ait.) sawdust was kindly supplied by the Portuguese company Valco–Madeiras e Derivados, S.A (Leiria, Portugal). The sawdust was initially sieved (mesh size of 20) and oven dried at 105 °C. LA (98 w/w%, MW = 116.12 g/mol and a density of 1.13 g/mL) was purchased from Acros Organics and FA (99 wt %, MW = 46.03 g/mol and a density of 1.22 g/mL) was purchased from CARLO ERBA. Lactic acid (90 w/w%, MW = 90.08 g/mol and density of 1.21 g/mL) and glacial acetic acid (99.9 w/w%, MW = 60.05 g/mol and density of 1.05 g/mL) were acquired from VWR Chemicals, while sulphuric acid (72%) was obtained from Chem-lab NV (MW = 98.08 g/mol and a density of 1.63 g/mL). De-ionized water was used for the preparation of all solutions. Dichloromethane (DCM) was purchased from Sigma-Aldrich and the 4-nitroanisol was acquired from Dagma. Microcrystalline cellulose Avicel PH-101, with an average particle size of 50 mm and degree of polymerization of ca. 260, was acquired from Sigma Aldrich. For the intrinsic measurements and molecular weight (MW) estimation of lignin, n,n-dimethylformamide (DMF) (≥99.8%) ACS Reagent, was used. 

### 2.2. Biomass Fractionation

The fractionation of pine sawdust was performed using different temperatures, extraction times and LA:FA ratios. In a typical experiment, the desired amount of biomass, ca. 1.5 g (dry basis), previously dried in an oven at 105 °C, was weighed and transferred to a metallic cylindric reactor, able to support high pressure. The reactor was filled with the solvent up to its maximum capacity (10 ml) and the vessel was properly closed. The reactor was placed in an oven, at the desired temperature (i.e., 120, 140, and 160 °C) and time (i.e., 1, 2, 4, and 6 h). After the end of the extraction, the vessel was carefully opened and the extracted lignin content estimated, as described below. 

### 2.3. Statistical Analysis

Extraction yields were calculated as the average of extracted residue in duplicate and total lignin recovery was determined based on the residue yield and lignin content [25]. 

Statistical analysis was performed using one-way ANOVA (α = 0.05) to evaluate significant differences between the extraction yields. 

### 2.4. Determination of Lignin Content

The lignin content extracted by the solvent was estimated using the standard LAP-004 protocol from the National Renewable Energy Laboratory (NREL) [29]. In brief, ca. 300 mg of the extract was weighed and hydrolyzed in 3 mL of 72 % of sulphuric acid solution (12 M) for 60 min at 30 °C, with intermittent stirring. Then, the hydrolysates were diluted to obtain a 4 % sulphuric acid solution, and autoclaved at 121 °C for 60 min, and left at room temperature to cool down. The autoclaved solutions were vacuum filtered using weighted filtering crucibles (40 mm diameter and porosity grade G2), to allow gravimetric determination of the “acid-insoluble lignin”, after washing the insoluble material. Aliquots of the filtrates were adequately diluted and used in the determination of the “acid-soluble lignin” by measuring the absorbance of the solution at 205 nm in a UV-VIS spectrometer (JASCO V650 Spectrophotometer). 

### 2.5. Viscosity Average Molecular Weight of Lignin

The average molecular weight (MW) of the extracted lignin was estimated from the intrinsic viscosity, according to the Huggins equation [30]. The intrinsic viscosity [*η*] can be related to the MW of the polymer by the semi-empirical Mark–Houwink equation [31],
$$\left[\eta \right]=KMw^{\alpha }$$
where *K* and α are constants specific for the determination conditions (in this case, *α* = 0.11; *K* = 2.51 [30]). The lignin was dissolved in DMF at a concentration of ca. 5 g/L. The solution was agitated for 2 h and allowed to stand for 24 h. Then, it was filtered with a 0.45/μm nylon syringe filter (filtraTech) to exclude any large particles from the solution. Subsequent lower concentrations, used for the [*η*] determination, were obtained by automatic volumetric dilution in a Viscologic TI1viscometer (Sematech, Nice, France).

### 2.6. Solvatochromic Kamlet–Taft Measurements

The solvatochromic probe 4-nitroanisol was used to estimate the Kamlet–Taft parameter π* (polarizability index). A stock solution of the dye in DCM was prepared to a concentration of 4 mM. The dye stock solution was added to the solvents understudy to a final concentration of 0.1 mM, which enables to obtain the absorbance values within the required measurable range, and DCM was removed by evaporation at room temperature. The absorption spectra of the solvatochromic probe were recorded from 250 nm to 500 nm in a quartz cell with a 10 mm path length, using a Shimadzu UV/VIS spectrometer UV-1700 at 1 nm stepwise. The wavelength at maximum absorption ν_max_ was determined for the probe in each solvent. The solvatochromic π* parameter was determined using the equation [32].
$$\pi^{*}=\;\frac{\left(34.12\;-\;\nu_{\max}\right)}{2.343}$$

The constant values were obtained from multiple correlation equations where the best fit for Δν_max_ was normalized to provide π* values for 28 solvents on a scale of 0 to 1, which was consistent with a π* of zero for cyclohexane, due to its low polarity, and unity for dimethyl sulfoxide, which is a highly polar organic liquid [33].

### 2.7. Scanning Electron Microscopy

The microscopic morphology of the pine sawdust particles was evaluated before and after the fractionation procedure, using a tungsten cathode scanning electron microscope SM 6010LV/6010LA, Jeol (Tokyo, Japan). Secondary electron mode, an acceleration voltage of 1 kV and a working distance of 9 mm, were selected as the operational conditions. Uncoated samples were deposited directly on the carbon tape. 

### 2.8. Fourier Transform Infrared (FTIR) Spectroscopy

FTIR-ATR spectra of the different samples were obtained on a JASCO FT/IR-4200 spectrometer (JASCO, Tokyo, Japan) using a MKII Golden Gate accessory. The spectra were recorded in the 500–4000 cm^−1^ range with a resolution of 4 cm^−1^ and 64 scans.

## 3. Results

### 3.1. Optimization of the Lignin Extraction Conditions

The raw lignocellulosic material used in this work was pinewood sawdust and its lignin content was initially estimated to be 27.36 ± 7.93%, which is in agreement with the literature [1,34,35].

As the extraction conditions are determinant for a successful fractionation process, it was intended to initially study the effect of the extraction time (i.e., 1, 2, 4 and 6 h) and temperature (i.e., 120, 140, and 160 °C) on the extraction yield, by systematically varying these two parameters for a fixed LA:FA solvent ratio of 1:1 (*v*/*v*) (Figure 1). 

Two striking observations can be made from the analysis of Figure 1: (1) the increase in temperature favors the extraction yield; (2) the content of extracted lignin also increases for longer extraction times. Nevertheless, above 4 h, no significant improvements are observed, and, in some cases, the extraction yield even decreases. This decrease in the extraction yield can be related to condensation reactions of lignin with reactive degradation products from hemicellulose, and consequent re-adsorption of these pseudo-lignin products on the surface of wood particles [36]. It is also noted that the greatest improvement in the extraction yield was obtained when the extraction time was extended to 2 h. 

Similarly, to other biomass residues, the pinewood sawdust consists not only of lignin but also of cellulose, hemicellulose, pectins, sugars and other minor compounds [29,37]. Often, during the fractionation process, the extracted lignin is contaminated with other compounds, particularly if the method used is poorly selective. To understand the selectivity of the novel LA:FA solvent system, the purity of the extracted fractions, obtained from the different pine fractionation conditions, was analyzed. Figure 2 shows the purity of the extracted fractions for 1:1 LA:FA solvent, at different temperatures and extraction times up to 6 h. As discussed above, the purity was not analyzed for extraction times longer than 6h since the extracted yield is not improved.

As it can be observed in Figure 2, the increase in temperature and extraction time apparently benefits the purity of the extracted lignin. The only exception is observed for the extraction performed at 160 °C and 4 h, where a drop in purity is observed. A slight fluctuation in purity of the extracted lignin for longer extraction times, was noticed. This is most likely due to dissolution of other wood compounds, such as hemicellulose, increasing the extraction yield, but decreasing the purity of the extracted lignin. In fact, the standard deviation obtained for longer extraction times is typically larger than for shorter extraction times. It is also noted that, at the lowest temperature, even after 4 h of extraction, the purity of the obtained lignin is quite unsatisfactory (below 50%). 

Similarly to the purity of the lignin, the MW is also influenced by the extraction time and temperature (Figure 3). The increase in temperature and extraction time generally causes the depolymerization of lignin (see arrows Figure 3). As can be observed in Figure 4, depolymerization is also inferred by the different colors of the extracted lignin, both the dried residue and in solution. Typically, the lower the MW, the darker the dried lignin powder and solution are.

Lignin is almost colorless in wood, while industrial lignin presents color due to the appearance of different chromophores during the extraction process. Lignin mainly includes quinoids, catechols, aromatic ketones, stilbenes, conjugated carbonyls with phenolics and metal complexes that are able to modify its coloration [38]. In addition, other chromophore groups may arise from leucochromophore oxidation and/or from carbohydrate contamination [39].

The LA:FA ratio was also analyzed (Figure 5) regarding its effect on the extraction yield and lignin purity, for the optimized extraction conditions (i.e., 160 °C and 4 h). 

All tested ratios induce enhanced lignin purity from ca. 62 to 83%, considerably above the 56% obtained for FA alone. Regarding the extraction yield, it remains fairly constant (around 29%) when the FA content in the solvent mixture varies between 50 to 90%. The highest extraction yields (around 40%) are obtained for the 7:3 and 6:4 ratios (LA:FA); however, the purity was compromised. Lignin with a lower MW is obtained when solely using FA as the extraction solvent. The lignin depolymerization strongly depends on the FA/LA ratio and, therefore, depending on the intended application, it is possible to obtain lignins with different MW without the need of using other depolymerization processes [40].

Overall, data suggests that a balance between the two acids is favorable for the pine sawdust fractionation, where enhanced extraction yields can be obtained with reasonable specificity, thus avoiding other expensive and complicated pre-treatments. Moreover, these two solvents appear to have different roles during the lignin extraction process but, when combined, their synergy results in a more efficient fragmentation; as a smaller molecule, FA is capable of penetrating into the sawdust structure and destabilize it, while LA seems to establish preferential interactions with lignin, via its extra ketone groups. 

### 3.2. Extraction Efficiency and Solvent Polarizability 

The effect of using other monocarboxylic acids in combination with LA on the extraction efficiency was also assessed. The results show a clear dependence of the extraction yield with the acid being used in combination with LA; lower extraction yields were obtained for the binary solvent systems in the following order: LA:FA > LA:lactic acid > LA:acetic acid (Figure 6a). The solvent efficiency regarding lignin extraction was correlated with their polarizability via the solvatochromic parameter π* (Figure 6b).

Data suggests that the lignin extraction is favored when using acidic solvents with higher π* values; LA and FA have the highest polarization indexes and, when combined to form a binary solvent system, were found more effective than LA:lactic acid or LA:acetic acid. These results are in accordance with our previous work where it was demonstrated that LA and formic acids have a similar and superior dissolution efficiency for “model” lignin (kraft lignin) and also the highest π* values, in comparison to other carboxylic acids [41]. A similar relation between π* and lignin solubility was also verified in alcohols, where π* of methanol was higher than that of 2-propanol [42] and the lignin was more soluble in the former alcohol. Our results reinforce the hypothesis that the polarizability of the solvent influences its ability to dissolve lignin. This can be an important parameter to consider when developing new solvent systems for lignin extraction/dissolution [43].

### 3.3. Scanning Electron Microscopy 

Scanning electron microscopy was used to observe the microscopic morphological features of pine sawdust, before and after fractionation with a solvent ratio mixture of 6:4 (LA:FA) at different extractions times and temperatures (Figure 7). The starting pine sawdust material exhibits a dense fibrous structure, where the characteristic “pits” of pinewood and some longitudinal fractures between the fibers can be observed, probably originated from the mechanical cut (Figure 7a). Depending on the extraction conditions, pores appear with different extent combined with “pits” (Figure 7b–d). In Figure 7d, the entire shape of the fibers is visible, suggesting a good fractionation of the biomass at 160 °C; while in Figure 7b,c, the fractionation of the biomass structure at 140 °C, is not so evident. At 160 °C, the fiber bundles were disrupted resulting in the removal of cross-linking structures between hemicelluloses and lignin, thus exposing the cellulose fibrils. These observations are in agreement with the extraction yields depicted in Figure 2, where the highest values were obtained at 160 °C with the solvent ratio of 6:4 (LA:FA). A careful observation suggests that the fractionation process leads to a decrease in the particle size and a slight browning of the recovered material. This can be associated with incomplete lignin extraction, with some residual lignin remaining in the cellulose-rich fraction or with the appearance of chromophores during the fractionation process [44]. 

### 3.4. Fourier Transform Infrared Spectroscopy

FTIR spectroscopy can provide reliable insights on structural and chemical features of lignocellulosic materials [45]. Therefore, to clarify the efficiency and specificity of the fractionation methods used, the presence/absence of lignin in the cellulose-rich solid residues was evaluated by FTIR and the spectra are depicted in Figure 8.

The characteristic fingerprints from functional groups of carbohydrates and lignin appear in the 1800–1200 cm^−1^ region while the OH and CH vibration modes are detected in the 3800–2700 cm^−1^ region [46]. The vibrational band at ca. 1530–1480 cm^−1^ is mostly attributed to the stretching of the C=C bonds, part of the aromatic skeletal of lignin molecules. Furthermore, the C=O stretching conjugated to the aromatic ring bending mode is assigned to the band at ca. 1720–1660 cm^−1^. Finally, the bands at 1490 and 1510 cm^−1^ can be assigned to the vibration modes of the aromatic rings in lignin [47]. The previously highlighted bands can also be identified in the starting raw material, as well as in the extracted fraction (lignin-rich portion). However, in the cellulose-rich material and in the microcrystalline cellulose spectrum, these bands are not visible, thus confirming the remarkable ability of the novel binary LA:FA solvent mixture to fully extract lignin from pine sawdust. 

## 4. Conclusions

Environmentally benign and selective methods for biomass fractionation, enabling the isolation and valorization of the different fractions, such as cellulose, hemicellulose, and lignin, have been targeted in much recent research. In this work, a novel binary solvent composed of bio-sourced and renewable FA and LA was developed and successfully applied to fractionate pine sawdust waste biomass. In comparison with the individual solvent systems, the appropriate tuning of temperature, extraction time and LA: FA ratio, allows the extraction of lignin with high yield and purity. Generally, higher temperatures and extraction times led to better results. Apart from the time–temperature balance, the solvent nature was sought for, further envisioning the industrial translation of the developed method. This study further suggests that LA has a good affinity with lignin, also proven by the higher MW obtained for lignin extracted solely with LA compared with the lignin extracted with only FA. Both the electron microscopy and FTIR data support the enhanced lignin extraction performance by the novel binary LA:FA mixtures developed. This eco-friendly and sustainable approach is potentially very interesting for future lignin valorization into novel biomaterials of added-value.

## Figures and Tables

**Figure 1 polymers-13-01090-f001:**
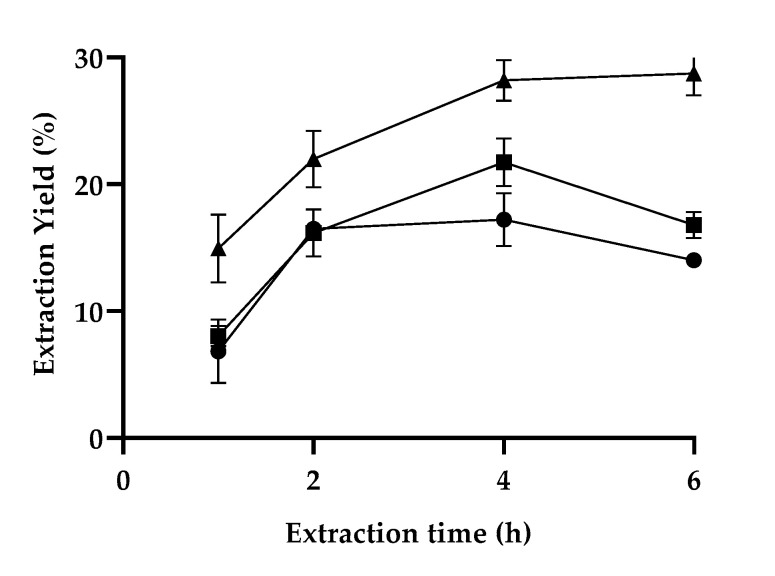
Extraction yield of LA:FA (1:1) as a function of extraction time for 120 °C (●), 140 °C (■) and 160 °C (▲).

**Figure 2 polymers-13-01090-f002:**
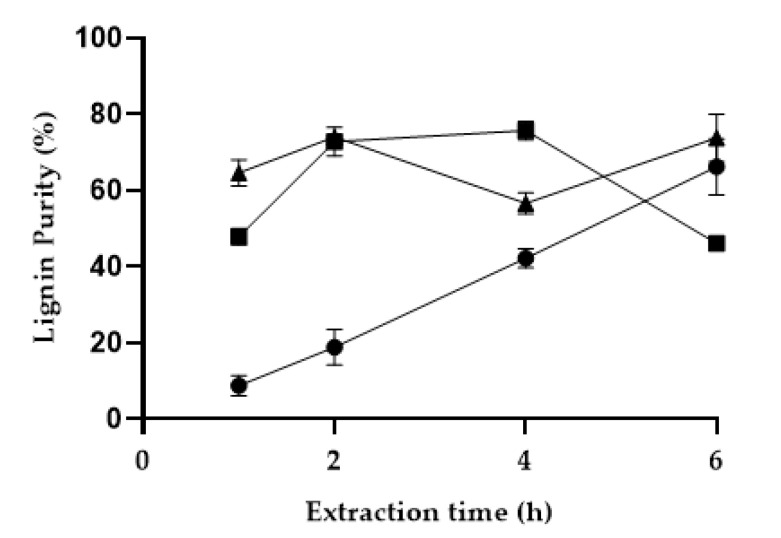
Purity of the lignin obtained by fractionation of the pine sawdust with the 1:1 LA:FA solvent system at different extraction times and temperatures, 120 °C (●), 140 °C (■) and 160 °C (▲).

**Figure 3 polymers-13-01090-f003:**
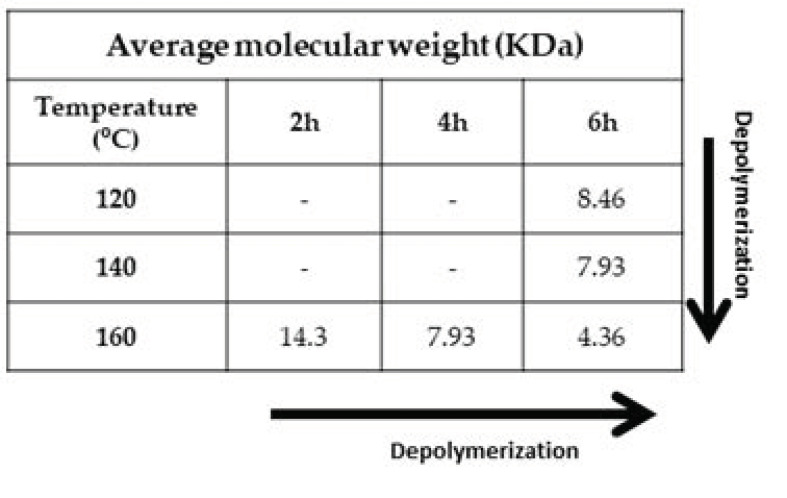
Average MW of the lignin obtained by fractionation of the pine sawdust with the 1:1 LA:FA solvent system at different extraction times and temperatures.

**Figure 4 polymers-13-01090-f004:**
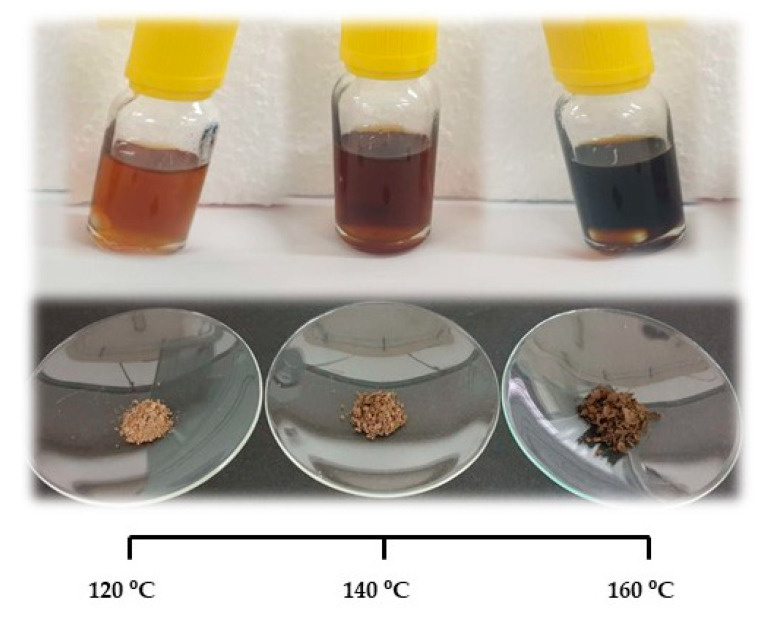
Visual appearance of the lignin powder and solution after being extracted with LA:FA (1:1) for 4 h at different temperatures.

**Figure 5 polymers-13-01090-f005:**
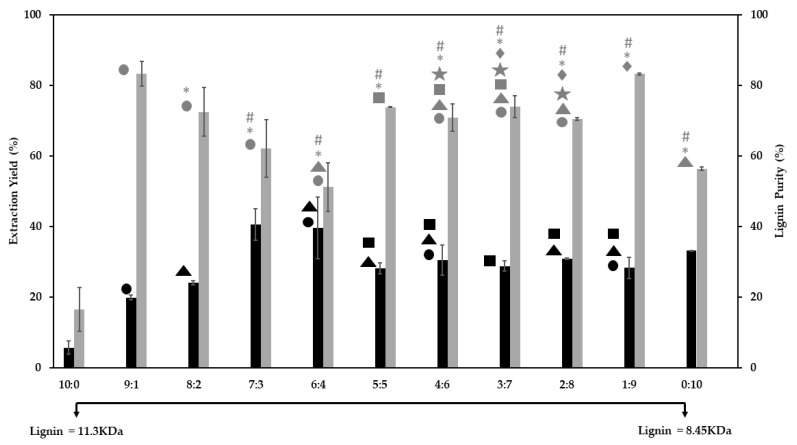
Extraction yield (black) and lignin purity (grey) obtained for different ratios of LA:FA, through the fractionation of pine sawdust at 160 °C for 4 h. The same symbols above the bars indicate no significant differences in the extraction yield and in the lignin purity from one-way ANOVA tests (*p* ≤ 0.05).

**Figure 6 polymers-13-01090-f006:**
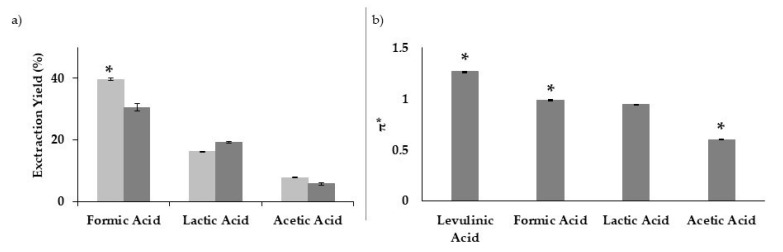
Extraction yield and polarization index π*: (**a**) effect of different acids on the extraction yield of solvent mixtures containing either 6:4 of LA:acid (dark grey) or 4:6 LA:acid (light grey), for 2 h at 160 °C; (**b**) π* values for different acids. The same symbols above the bars indicate no significant differences in the extraction yield and in the lignin purity (Figure 6a) and π* values (Figure 6b) from one-way ANOVA tests (*p* ≤ 0.05).

**Figure 7 polymers-13-01090-f007:**
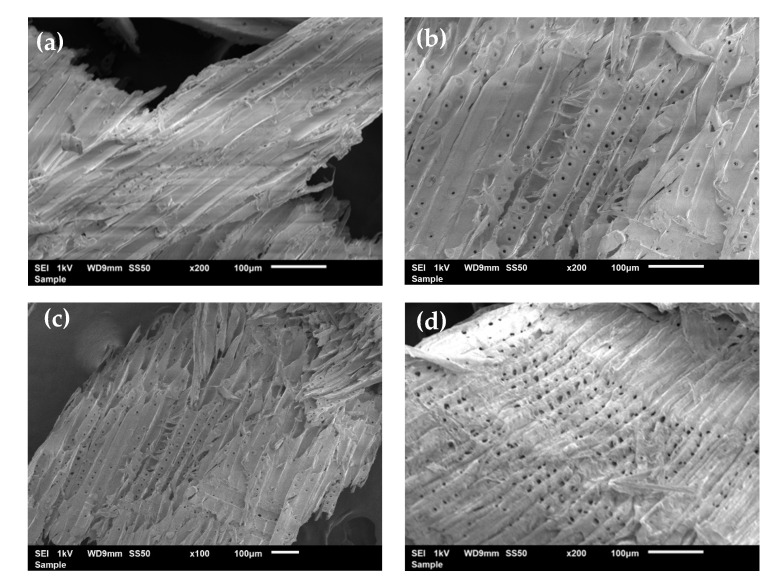
Scanning electron microscopy of the starting pine sawdust (**a**), and the cellulose-rich materials obtained after fractionation with 6:4 LA:FA lasting (**b**) 2 h at 160 °C, (**c**) 2 h at 120 °C, and (**d**) 4 h 160 °C.

**Figure 8 polymers-13-01090-f008:**
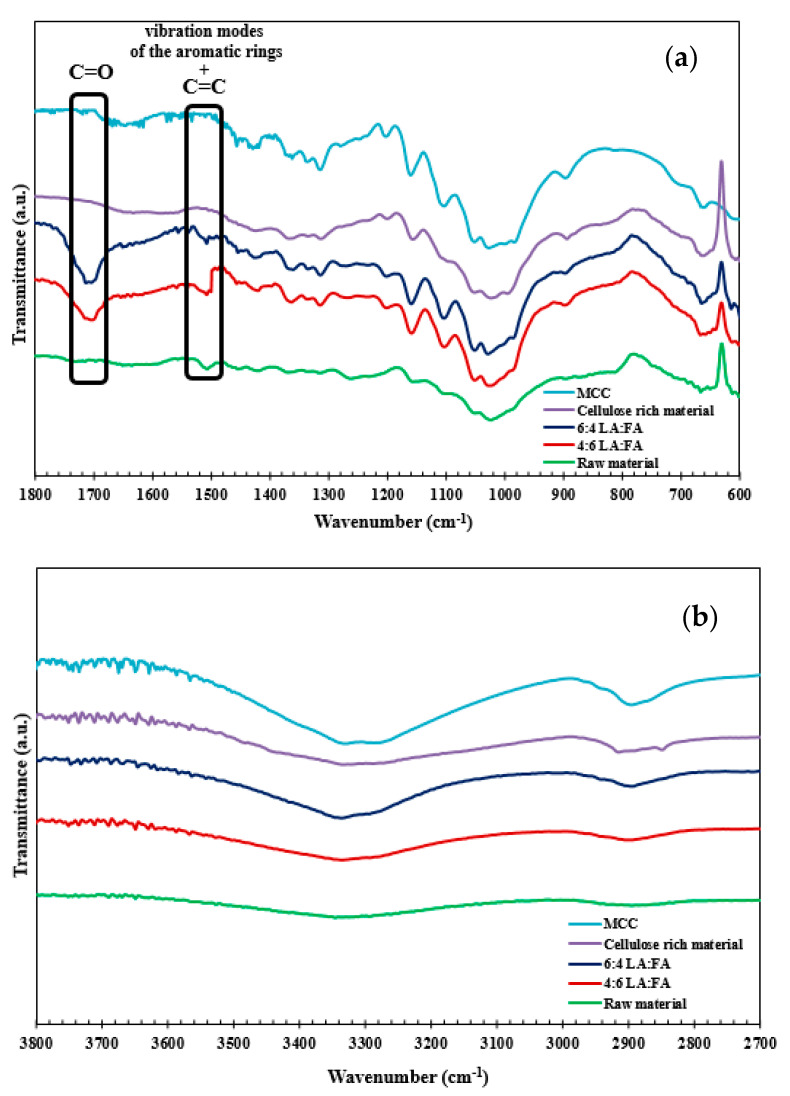
FTIR spectra for low (**a**) and high (**b**) wavenumbers of pre-extracted pine sawdust (raw material), the lignin obtained from the fractionation with 4:6 LA:FA (4 h at 160 °C), the lignin obtained from the fractionation with 6:4 LA:FA (4 h at 160 °C), cellulose-rich material obtained from the fractionation of pine sawdust with 6:4 LA:FA and “model” microcrystalline cellulose (MCC) from a commercial supplier.

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
