# Peer review of "Lignin Extraction from Waste Pine Sawdust Using a Biomass Derived Binary Solvent System"

_polymers, 2021, doi:10.3390/polym13071090_

Round 1

Reviewer 1 Report

This paper focused on the extraction of Lignin from waste pine sawdust using a biomass derived binary solvent system. The presented research work is an interesting and comprehensive research work and it is useful for production of added-value products, represents an important opportunity to reduce and valorize agroforest residues. I would recommend this manuscript for acceptance after the points listed below are addressed.

In abstract authors should mention the importance of research work briefly.

Introduction should be substantially improved. To highlight the significance of the study, author should introduce the development of production of added-value products, especially the extraction of active substances from wastes or by-products, which can reduce and valorize agroforest residues and be contributed to establish a circular economy to minimize resource use and waste. Some very recent studies should be mentioned as examples.

Reutilization of food waste: One-step extration, purification and characterization of ovalbumin from salted egg white by aqueous two-phase flotation. Foods 2019, 8, 286. doi:10.3390/foods8080286

Effective separation of prolyl endopeptidase from Aspergillus Niger by aqueous two phase system and its characterization and application. Int. J. Biol. Macromol. 2021, 169, 384–395. doi: 10.1016/j.ijbiomac.2020.12.120

Direct separation and purification of α-Lactalbumin from cow milk whey by aqueous two-phase flotation of thermo-sensitive polymer/phosphate. J. Sci. Food Agric. 2021. doi: 10.1002/jsfa.11097

Fabrication and characterization of a microemulsion stabilized by integrated phosvitin and gallic acid. J. Agric. Food Chem. 2020, 68, 5437–5447. doi: 10.1021/acs.jafc.0c00945

The introduction about lignin can be shorten because lignin is already well understood.

Biomass fractionation: Were stirring performed during the extraction process?

Line 157: K should be italic.

Figure 2: Why the extraction yield decreased above 4 h in case of 120 ºC and 140 ºC?

The method of purity determination should be described in section Materials and Methods.

Reasons for fluctuations in purity should be discussed.

Figure 6: The filling method of the graph should be modified to see the error bars clearly

In general, the discussion should be heavily improved.

Author Response

Replies to reviewer’s comments.

#Reviewer 1

This paper focused on the extraction of Lignin from waste pine sawdust using a biomass derived binary solvent system. The presented research work is an interesting and comprehensive research work and it is useful for production of added-value products, represents an important opportunity to reduce and valorize agroforest residues. I would recommend this manuscript for acceptance after the points listed below are addressed.

In abstract authors should mention the importance of research work briefly.

Reply: We thank the reviewer for recognizing the importance of our work. We believe we have clearly stressed the importance of our work in the abstract by stating that “…the novel combination of bio-sourced FA and LA has shown to be a very promising system for lignin extraction of high purity from biomass waste, thus contributing to extend the opportunities of lignin manipulation and valorisation into novel added-value biomaterials”.

Introduction should be substantially improved. To highlight the significance of the study, author should introduce the development of production of added-value products, especially the extraction of active substances from wastes or by-products, which can reduce and valorize agroforest residues and be contributed to establish a circular economy to minimize resource use and waste. Some very recent studies should be mentioned as examples.

Reutilization of food waste: One-step extration, purification and characterization of ovalbumin from salted egg white by aqueous two-phase flotation. Foods 2019, 8, 286. doi:10.3390/foods8080286

Effective separation of prolyl endopeptidase from Aspergillus Niger by aqueous two phase system and its characterization and application. Int. J. Biol. Macromol. 2021, 169, 384–395. doi: 10.1016/j.ijbiomac.2020.12.120

Direct separation and purification of α-Lactalbumin from cow milk whey by aqueous two-phase flotation of thermo-sensitive polymer/phosphate. J. Sci. Food Agric. 2021. doi: 10.1002/jsfa.11097

Fabrication and characterization of a microemulsion stabilized by integrated phosvitin and gallic acid. J. Agric. Food Chem. 2020, 68, 5437–5447. doi: 10.1021/acs.jafc.0c00945

Reply: We thank the reviewer suggestion. The introduction was improved highlighting some applications regarding lignin-based added-value products.

The introduction about lignin can be shorten because lignin is already well understood.

Reply: The introduction was revised.

Biomass fractionation: Were stirring performed during the extraction process?

Reply: No, our fractionation system does not allow stirring during the extraction process. Indeed, stirring could allow a better dispersion and interaction of the solvent system with the pine sawdust, thus potentially improving the extraction yields. We hope in the future to modify the reactor used and explore the stirring effect on the fractionation process.

Line 157: K should be italic.

Reply: Done.

Figure 2: Why the extraction yield decreased above 4 h in case of 120 ºC and 140 ºC?

Reply: The decrease in the extraction yield may be related with condensation reactions of lignin with reactive degradation products of hemicellulose, and consequent re-adsorption of these pseudo-lignin products on the wood particles surface. This hypothesis was added to manuscript.

The method of purity determination should be described in section Materials and Methods.

Reply: The method was described in Materials and methods section “Determination of lignin content”.

Reasons for fluctuations in purity should be discussed.

Reply: A slight fluctuation in purity of the extracted lignin, especially observed for longer extraction times, is most likely related with the co-dissolution of other wood compounds, such as hemicellulose, which not only contributes to increase the extraction yield, but also decreases the purity of the extract in lignin. In fact, the standard deviation obtained for longer extraction times is typically larger than for shorter extraction times. This discussion was added to manuscript.

Figure 6: The filling method of the graph should be modified to see the error bars clearly

Reply: Done.

In general, the discussion should be heavily improved.

Reply: The discussion of the results was revised and improved accordingly.

Reviewer 2 Report

In the manuscript polymers-1141466 a novel lignin extraction process is discussed, by using bio-based and renewable chemicals, such as a mixture from levulinic acid and formic acid, in order to fractionate lignocellulosic waste (i.e., Maritime pine sawdust) and isolate lignin.

In my opinion, the manuscript needs to be improved and I recommend the publication in Polymers journal after major revisions:

  1. Introduction

- The representation of the main structural units of lignin (Figure 1) it is very known to the audience and therefore I propose its removing!

- p. 3, L. 92-105: The main objectives of the manuscript are not well emphasized among the literature information! Please reformulate this paragraph and try to evidence the purpose of the manuscript!

- p. 4, L. 107: Please specify the type of electron microscopy which was used in the manuscript, in order to avoid a confusion between SEM and TEM!

  1. Materials and Methods

Materials

- p. 4, L. 110: Please add some characteristics for Maritime pine (Pinus pinaster Ait.) sawdust, like size of sawdust particles, the preparation method, etc.

- p. 4, L. 111: “Levulinic acid” not “Levulinc acid”! Please make the correction!

- p. 4, L. 111-113: Please add the abbreviation LA and FA also in Materials section!

Biomass fractionation

- p. 4, L. 129: Please mention the all the temperatures and rection time used in the experiment.

Statistical analysis

- p. 4, L. 134: I would like to know how the authors determined the purity of lignin? If the authors used a standard procedure (as it is mentioned in their own paper [24]), please add the correct reference! Moreover, in my opinion the authors determined the lignin content present in pine sawdust and not the lignin purity. Please revise the paragraph!

Determination of lignin content

- p. 5, L. 142: “300 mg of the extract was weighed and hydrolyzed in 3 mL of 72 % of sulphuric acid solution (12 M) for 60 min at 30 ºC. The reaction was allowed to proceed for more 60 min with intermittent stirring”! Please correct the reaction time, if is 60 min or more than 60 min!

- p. 5, L. 145: What type of weighted filtering crucibles did the authors used? Please add the information!

3. Results

3.1. Optimization of the lignin extraction conditions

- p. 6, L. 196-199: Please mention the temperatures and times used in this experiment for a better understanding!

- p. 7, L. 219-220: Figure 3b it is actually a table! Please make it to looks as a figure, or mention it separately as a table!

- p. 9, L. 243: Please add at figure caption the legend of all symbols!

3.2. Extraction efficiency and solvent polarizability

- p. 10, L. 283: There are two full stops!

3.3. Scanning electron microscopy

- p. 10, L. 294: Please explain why the authors chose to study the fractions obtained after a fractionation with a solvent ratio mixture of 6:4 (LA:FA)!

- p. 11, L. 299-301: Please change “panel” with” Figure7” and corresponding letter!

- p. 11, L. 310-311. Please remove the photographs from Figure 7 (Figure 7e and 7f)! In this section is discussed the morphology of the samples observed by SEM microscopy! Moreover, the information from L. 305-309 is already mentioned at Section 3.1. If the authors want to maintain these figures please move them to Section 3.1!

- p. 11, L. 310: Please explain why appear at figure caption the sample “c) 8 h at 140 °C”? The authors mentioned that “the purity was not analyzed for extraction times longer than 6h since the extracted yield is improved”!

- p. 11, L. 310: It would be more interesting to add a SEM image for sample “f) 4 h at 160 °C”, if this is considered the optimized extraction conditions!

The Section 3.3 needs to be improved and the most significant figures need to be added, in correlation with the data obtained.

3.4. Fourier transform infrared spectroscopy

- In Figure 8 the reader can not identify the characteristic bands discussed in Section 3.4! Please add the characteristic bands or make it visible!

- The finger print region brings important information which cannot be observed in Figure 8, and thus, the discussed modifications cannot be identify! Please make two different figures for each important region (3800-2700 cm-1 and 1800-600 cm-1).

Section 3.4 is poorly explained, with brief information, without going into detail. Please revise the entire FTIR section!

Author Response

Replies to reviewer’s comments.

#Reviewer 2

In the manuscript polymers-1141466 a novel lignin extraction process is discussed, by using bio-based and renewable chemicals, such as a mixture from levulinic acid and formic acid, in order to fractionate lignocellulosic waste (i.e., Maritime pine sawdust) and isolate lignin.

In my opinion, the manuscript needs to be improved and I recommend the publication in Polymers journal after major revisions:

Reply: We thank the reviewer for the comments on our work. 

  1. Introduction

- The representation of the main structural units of lignin (Figure 1) it is very known to the audience and therefore I propose its removing!

Reply: The reviewer suggestion was followed, and the figure was removed. 

- p. 3, L. 92-105: The main objectives of the manuscript are not well emphasized among the literature information! Please reformulate this paragraph and try to evidence the purpose of the manuscript!

Reply: The mentioned paragraph was reformulated according to the reviewer suggestion.

- p. 4, L. 107: Please specify the type of electron microscopy which was used in the manuscript, in order to avoid a confusion between SEM and TEM!

Reply: Done.

  1. Materials and Methods

Materials

- p. 4, L. 110: Please add some characteristics for Maritime pine (Pinus pinaster Ait.) sawdust, like size of sawdust particles, the preparation method, etc.

Reply: Additional characteristics of Maritime pine sawdust were added to manuscript.

- p. 4, L. 111: “Levulinic acid” not “Levulinc acid”! Please make the correction!

Reply: Done.

- p. 4, L. 111-113: Please add the abbreviation LA and FA also in Materials section!

Reply: The abbreviations were added to the materials section.

Biomass fractionation

- p. 4, L. 129: Please mention the all the temperatures and rection time used in the experiment.

Reply: The temperatures and reaction times were added.

Statistical analysis

- p. 4, L. 134: I would like to know how the authors determined the purity of lignin? If the authors used a standard procedure (as it is mentioned in their own paper [24]), please add the correct reference! Moreover, in my opinion the authors determined the lignin content present in pine sawdust and not the lignin purity. Please revise the paragraph!

Reply: We appreciate the reviewer comment. The paragraph has been revised accordingly.

Determination of lignin content

- p. 5, L. 142: “300 mg of the extract was weighed and hydrolyzed in 3 mL of 72 % of sulphuric acid solution (12 M) for 60 min at 30 ºC. The reaction was allowed to proceed for more 60 min with intermittent stirring”! Please correct the reaction time, if is 60 min or more than 60 min!

Reply: The reaction time was corrected. The reaction time is 60 min.

- p. 5, L. 145: What type of weighted filtering crucibles did the authors used? Please add the information!

Reply: The weighted filtering crucibles consist of crucibles with 40mm diameter and a G2 porous size. This information was added to manuscript.

  1. Results

3.1. Optimization of the lignin extraction conditions

- p. 6, L. 196-199: Please mention the temperatures and times used in this experiment for a better understanding!

Reply: The temperatures and times were added.

- p. 7, L. 219-220: Figure 3b it is actually a table! Please make it to looks as a figure, or mention it separately as a table!

Reply: We agree with the reviewer comments and Figure 3 has been divided in a figure (Figure 2) and a table (Table 1).

- p. 9, L. 243: Please add at figure caption the legend of all symbols!

 Reply: The symbols above the bars were added to identify the ratios between LA and FA which represent, or not, significant differences in the extraction yield. Thus, it is not reasonable to attribute a “legend” to each symbol. As mentioned in the figure caption: “The same symbols above the bars indicate no significant differences in the extraction yield and in the lignin purity from One-Way Anova test (P≤0.05).”

3.2. Extraction efficiency and solvent polarizability

- p. 10, L. 283: There are two full stops!

 Reply: Corrected.

3.3. Scanning electron microscopy

- p. 10, L. 294: Please explain why the authors chose to study the fractions obtained after a fractionation with a solvent ratio mixture of 6:4 (LA:FA)!

Reply: The 6:4 ratio is one of the solvent mixtures that gave high extraction yield. However, the lignin content (purity) was compromised. Thus, we have chosen this ratio to understand if the changes in morphology of the pine sawdust can explain the low lignin content; it was observed that at 160ºC the morphology strikingly changed revealing that, most likely, not only lignin was dissolved but also some hemicellulose, leading to high extraction yield but low lignin purity.

- p. 11, L. 299-301: Please change “panel” with” Figure7” and corresponding letter!

Reply: Done.

- p. 11, L. 310-311. Please remove the photographs from Figure 7 (Figure 7e and 7f)! In this section is discussed the morphology of the samples observed by SEM microscopy! Moreover, the information from L. 305-309 is already mentioned at Section 3.1. If the authors want to maintain these figures please move them to Section 3.1!

Reply: We agree with the reviewer and the photographs were moved to supplementary material (Figure S1).

- p. 11, L. 310: Please explain why appear at figure caption the sample “c) 8 h at 140 °C”? The authors mentioned that “the purity was not analyzed for extraction times longer than 6h since the extracted yield is improved”!

Reply: We apologize for the mistake. We have added the image obtained for the sample fractionated during 2h at 120ºC.

- p. 11, L. 310: It would be more interesting to add a SEM image for sample “f) 4 h at 160 °C”, if this is considered the optimized extraction conditions!

Reply: We apologise for the mistake in the figure legend and thank the reviewer for noticing it. In fact, the previous Figure 7d (now Figure 6d) is the image of the sample treated at 160 ºC during 4h. Captions have been corrected.

The Section 3.3 needs to be improved and the most significant figures need to be added, in correlation with the data obtained.

Reply: Section 3.3. has been revised. The photographs have been removed and the captions corrected.

3.4. Fourier transform infrared spectroscopy

- In Figure 8 the reader can not identify the characteristic bands discussed in Section 3.4! Please add the characteristic bands or make it visible!

Reply: The FTIR spectra were replotted in a different way to highlight the bands of interest.

- The finger print region brings important information which cannot be observed in Figure 8, and thus, the discussed modifications cannot be identify! Please make two different figures for each important region (3800-2700 cm-1 and 1800-600 cm-1).

Reply: As suggested by the reviewer, the previous Figure 8 (now Figure 7) was divided in two different figures to highlight the regions of interest.

Section 3.4 is poorly explained, with brief information, without going into detail. Please revise the entire FTIR section!

Reply: The spectra have been replotted and the major bands have been clearly identified with support from suitable literature. Therefore, we believe we have provided sufficient information and discussion regarding the FTIR data. It is not clear what further information the reviewer would like to see explored.
